# Development of innovative alkali activated paste reinforced with polyethylene fibers for concrete crack repair

Munir Iqbal[1]*, Muhammad Ashraf[1], Sohaib Nazar[2¤], Loai Alkhattabi[3], Jihad Alam[1], Hisham Alabduljabbar[4], Zahoor Khan[1]

1 Department of Civil Engineering, Ghulam Ishaq Khan Institute of Engineering Sciences and Technology, Topi, Pakistan, 2 Shanghai Key Laboratory for Digital Maintenance of Buildings and Infrastructure, School of Naval Architecture, Ocean and Civil Engineering, Shanghai Jiao Tong University, Shanghai, China, 3 Department of Civil and Environmental Engineering, College of Engineering, University of Jeddah, Jeddah, Saudi Arabia, 4 Department of Civil Engineering, College of Engineering in Al-Kharj, Prince Sattam Bin Abdulaziz University, Al-Kharj, Saudi Arabia

¤ Current address: Department of Civil Engineering, Comsats University Islamabad-Abbottabad Campus, Pakistan
* muniriqbal0345@gmail.com

**Data Availability Statement:** Data is presented in the paper in form of graphs.

**Funding:** The author(s) received no specific funding for this work.

## Abstract

Concrete structures are susceptible to cracking, which can compromise their integrity and durability. Repairing them with ordinary Portland cement (OPC) paste causes shrinkage cracks to appear in the repaired surface. Alkali-activated binders offer a promising solution for repairing such cracks. This study aims to develop an alkali-activated paste (AAP) and investigate its effectiveness in repairing concrete cracks. AAPs, featuring varying percentages (0.5%, 0.75%, 1%, 1.25%, 1.5%, and 1.75%) of polyethylene (PE) fibers, are found to exhibit characteristics such as strain hardening, multiple plane cracking in tension and flexure tests, and stress-strain softening in compression tests. AAP without PE fibers experienced catastrophic failure in tension and flexure, preventing the determination of its stress-strain relationship. Notably, AAPs with 1.25% PE fibers demonstrated the highest tensile and flexural strength, exceeding that of 0.5% PE fiber reinforced AAP by 100% in tension and 70% in flexure. While 1% PE fibers resulted in the highest compressive strength, surpassing AAP without fibers by 17%. To evaluate the repair performance of AAP, OPC cubes were cast with pre-formed cracks. These cracks were induced by placing steel plates during casting and were designed to be full and half-length with widths of 1.5 mm and 3 mm. AAP both with and without PE fibers led to a substantial improvement in compressive strength, reducing the initial strength loss of 30%-50% before repair to a diminished range of 2%-20% post-repair. The impact of PE fiber content on the compressive strength of repaired OPC cube is marginal, providing more flexibility in using AAP with any fiber percentage while still achieving effective concrete crack repair. Considering economic and environmental factors, along with observed mechanical enhancements, AAPs show promising potential for widespread use in concrete repair and related applications, contributing valuable insights to the field of sustainable construction materials.

**Competing interests:** The authors have declared that no competing interests exist.

## Introduction

Ordinary Portland cement (OPC) is widely used in construction, with an annual global production of 4.0 billion tons and a growth rate of 4% per annum. However, the production process of cement leads to a substantial release of carbon dioxide ($CO_2$) into the atmosphere, primarily through limestone calcination and fuel combustion [1–3]. Each ton of cement produced emits one ton of $CO_2$, contributing to 7% of greenhouse gas emissions [4, 5]. In response to the need for mitigating greenhouse gas emissions, numerous studies have explored alkali activated binders (AABs) as alternatives to OPC. These composites, formed through the chemical reaction of alkali metal and silicate particles, offer a three-dimensional amorphous alumino-silicate matrix. The characteristics of AABs are influenced by several factors, including mixture rheology, desired compressive and flexural strength, modulus of elasticity, flexural toughness, ductility index, and shrinkage during drying [6, 7]. They possess durability, acid resistance, fire resistance, and good thermal insulation properties [8, 9]. AABs exhibit superior compressive strength [10], yet they face challenges due to low tensile strength and limited ultimate strain. Incorporating fibers into AABs presents a promising avenue for enhancing ductility and tensile strength. Fiber reinforcement, as seen in engineered cementitious composites (ECC) and engineered geopolymer composites (EGC), has proven effective in improving mechanical and post-crack properties [11, 12].

Various fiber types, such as steel, glass, basalt, and polyvinyl alcohol (PVA), have been utilized in fiber-reinforced concrete (FRC) [13–16]. The types of fibers, each having its own characteristic properties, significantly impacts performance such as tensile strength, ductility index, and post-cracking behavior [17]. Basalt fiber [18] increases compressive strength while steel fibers improve tensile strength and provide good freeze-thaw resistance. However, the later type present challenges in cost, workability, and durability [19–22]. The issue with glass fibers is their instability in alkaline environments [23]. PVA and polyethylene (PE) fibers are the most appropriate fibers for enhancing the overall concrete performance because of their stability in an alkaline environment and higher values of elastic modulus [24–26]. Although PE fibers decrease the slump, which is further aggravated by uneven fiber distribution, they enhance abrasion resistance and reduce drying shrinkage by more than 10% [27–30]. Moreover, the strength gained for PE fibers is greater than glass, synthetic, and PVA fibers [31, 32]. The length of PE fibers is crucial for tensile strain capacity and strain-hardening behavior [33]. The combination of PE and steel fibers in cementitious materials results in improved energy dissipation and good deformation behavior, thus enhancing the mechanical properties of cementitious composites in construction applications [27, 28, 33, 34].

Concrete structures commonly experience cracking due to factors like volumetric changes, thermal stresses, chemical reactions, reinforcement corrosion, poor tensile strength, and construction techniques [35–37]. Sulfate attack and chloride intrusion also pose risk to cementitious materials [38], putting buildings in danger. When concrete is loaded, cracks frequently form and spread which provides easy access to water and accelerates structural deterioration. To tackle these issues, diverse materials and methodologies have been utilized in the repair of concrete cracks [39, 40], with the selection process contingent on specific objectives, including restoring or enhancing stiffness, improving durability, increasing strength, enhancing appearance, ensuring water tightness, preventing corrosive environments, and improving overall functional performance. Reference guidelines from the American Concrete Institute, Eurocode and other entities are available for various concrete crack repair methods [ACI 224.1R-07, ISO/TC 71/SC 7, 1504–9, HB 84–2006] [40–43]. The width of the crack dictates repair materials and techniques, with options ranging from crack injection to gravity filling. Flexible materials like polysulfides or polyurethanes are recommended for live cracks, while polymer-

**Table 1. Chemical composition of fly ash.**

| Compound | SiO$_2$ | Al$_2$O$_3$ | Fe$_2$O$_3$ | CaO | Na$_2$O | MgO | P$_2$O$_5$ | SO$_3$ | TiO$_3$ | MnO | LOI |
|---|---|---|---|---|---|---|---|---|---|---|---|
| Fly Ash | 50.11 | 26.56 | 11.40 | 0.69 | 0.72 | 1.45 | 0.885 | 0.24 | 1.32 | 0.15 | 0.57 |

modified cementitious grouts are effective for wider cracks [39]. Crack filling or injection method involves enlarging the crack, then filling and sealing with appropriate joint sealants such as asphaltic materials, urethanes, polymer mortars, epoxies, polysulfides, and silicones. Rodler and Fulton [44, 45] outline gravity filling as a method employing low viscosity resins or monomers like urethanes and epoxies for repairing narrow cracks (0.03 mm to 2 mm). Cement mortar also serves as a prevalent crack repair material; however, its efficacy may be limited over the long term due to potential shrinkage cracks which causes the development of new cracks at the same location [10]. The gravity-fill technique utilizes low-viscosity resins like epoxy or high molecular weight methacrylate [46], which restore mechanical properties but may lead to failures at the epoxy-concrete interface under fatigue stress [47, 48].

As from the discussions above, it is clear that various materials, each with their own characteristic properties, have been tested for repair purposes. However, there are limited studies available on the utilization of AAPs as repair materials. In the first phase of this study, mechanical properties of AAPs reinforced with different proportions of PE fibers are assessed. The second phase focuses on the applicability of the developed AAPs as filler materials for the repair of 1.5 mm and 3 mm concrete cracks.

## Materials

### Class-F fly ash

The class-F fly ash used in this study complies with the ASTM Specification C 618 and is derived from anthracite coal [49]. It is obtained by separating combustion gases using mechanical precipitators in coal power plants. This type of fly ash is not self-cementitious and has a calcium content of less than 5%. The chemical composition of fly ash is evaluated using X-ray fluorescence (XRF) and is presented in Table 1. Fig 1 illustrates the particle size distribution of class F fly ash.

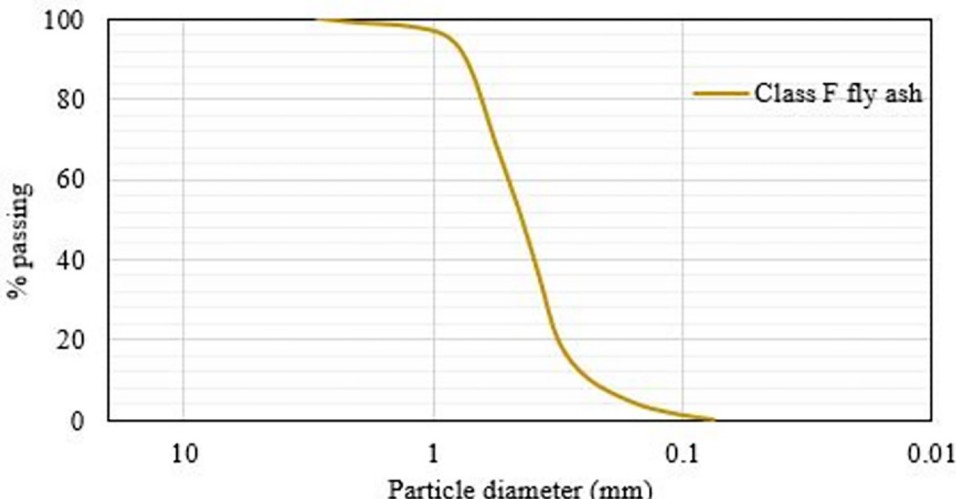

**Fig 1. Particle size distribution graph of class F fly ash.**

**Table 2. Composition of GGBS.**

| Composition | CaO | $SiO_2$ | $Al_2O_3$ | $Na_2O$ | KO | MgO | LOI |
|---|---|---|---|---|---|---|---|
| (%) | 40 | 42.3 | 12 | 0.95 | 0.65 | 1.4 | 2.2 |

**Table 3. Properties of PE fibers.**

| Fiber type | Length (mm) | Diameter (mm) | Elastic Modulus (MPa) | tensile strength (MPa) | Density ($gm/cm^3$) | Elongation (%) |
|---|---|---|---|---|---|---|
| PE | 12 | 0.012 | 73000 | 2580 | 0.97 | – |

### Ground granulated blast furnace slag (GGBS)

This study utilized GGBS as 30% of the total binder. Table 2 presents the composition of GGBS as obtained from the provider. The specific gravity of the GGBS is found to be 2.90, while its water absorption is 1.38%.

### Polyethylene fibers

PE is fatigue and wear-resistant plastics used to provide flexibility and high ductility. PE fibers used in this study have a higher tensile strength than PVA fibers [11]. Table 3 displays the characteristics of PE fibers, as provided by the manufacturer.

### Activators

Activators play a significant role in the polymerization process, which involves the transformation of precursor materials into a solid polymeric structure. The alkaline activator employed in this investigation consisted of 97% pure sodium hydroxide (NaOH) pellets and sodium silicate powder ($Na_2SiO_3$) with a specific gravity of 1.35. Further, the ratio of $SiO_2$ and $Na_2O$ is 3.2 in $Na_2SiO_3$. The materials used for the AAP formulation are depicted in Fig 2.

## Experimental procedure and methodology

To study the mechanical properties of AAP, several samples with different PE fiber percentages were cast for compression, tension, and flexural test. After being cast, the samples were kept in oven at 60˚C for 7 days and then left to cure at room temperature for 21 days.

### Mix proportions for AAP

The total binder utilized in this study consists of 30% GGBS and 70% class F fly ash. Water-to-binder (w/b) ratio is varied for the required flow and a small quantity (1–2% of the total binder) of superplasticizer (SP) is added. The activators added are 10% of the total binder, with NaOH accounting for 30% of the activator and $Na_2SiO_3$ accounting for the remaining 70%. This mix design is carried out with various percentages (0.5, 0.75, 1, 1.25, 1.5, and 1.75%) of PE fibers in the mix. All percentages are taken with respect to weight of the total binder (fly ash + GGBS). The above mix design and fiber dosages are taken from the extensive literature available on alkali activated mixes [50, 51]. Table 4 summarizes the mix proportions used in this study.

### Mixing procedure and sample preparation

In this study, the mixing procedure was based on the methodologies used by Nazar et al. and Yousefi et al. [52, 53]. The process involved several steps. Firstly, fly ash, GGBS, PE fibers, and

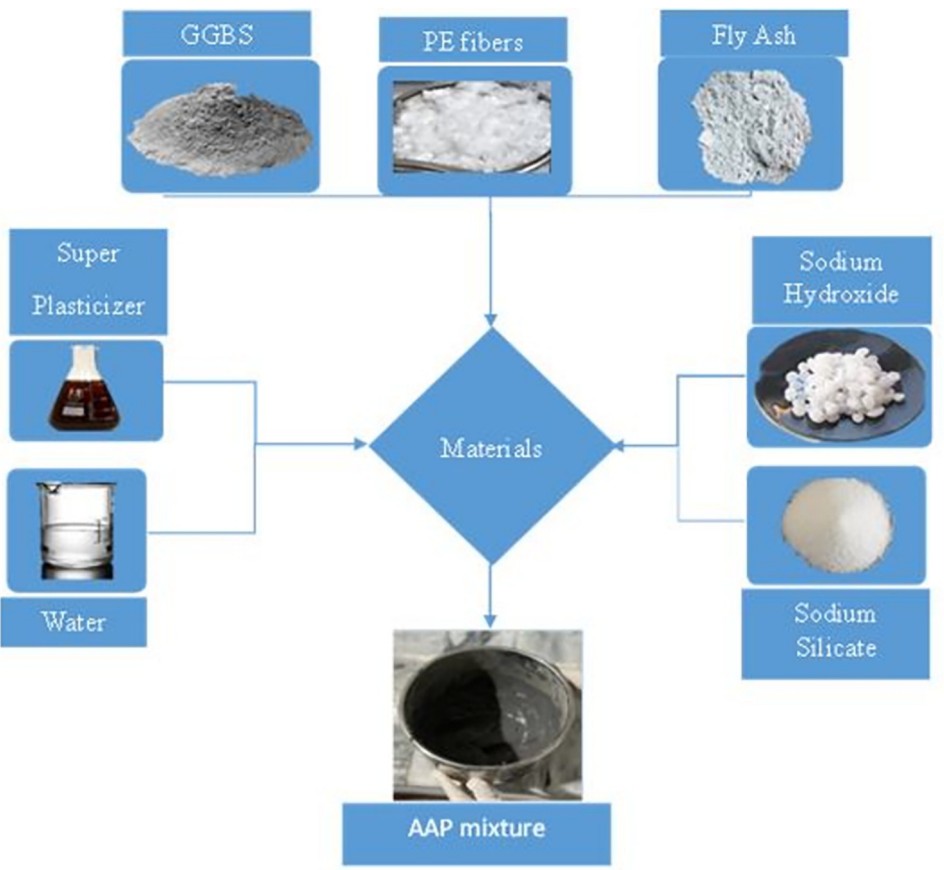

**Fig 2. Constituents of AAP.**

grinded alkali activators underwent a dry mixing process in a Hobart mixer for 2 minutes to ensure a uniform blending. Following this, water and a SP were added at low speed for 1 minute. Finally, fast mixing for 2 minutes was conducted to achieve a uniform mix slurry. Samples were prepared from this slurry. Rectangular beams of 320 mm length x 40 mm height x 12 mm thickness were cast for flexural test. The flexural strengths of the AAP mixes were tested at 28 days by using the method specified by ASTM C 293 [54].

**Table 4. Mix proportion of AAP.**

| Mix | GGBS (kg/m$^3$) | Fly Ash (kg/m$^3$) | PE fiber (kg/m$^3$) | Na$_2$SiO$_3$ (kg/m$^3$) | NaOH (kg/m$^3$) | W (kg/m$^3$) | SP (kg/m$^3$) |
|---|---|---|---|---|---|---|---|
| PE 0% | 460 | 1009 | 0 | 101 | 43 | 540 | 6 |
| PE 0.50% | 460 | 1009 | 7 | 101 | 43 | 555 | 9 |
| PE 0.75% | 460 | 1009 | 11 | 101 | 43 | 565 | 9 |
| PE 1.00% | 460 | 1009 | 14 | 101 | 43 | 576 | 9 |
| PE 1.25% | 460 | 1009 | 18 | 101 | 43 | 587 | 9 |
| PE 1.50% | 460 | 1009 | 21 | 101 | 43 | 600 | 14 |
| PE 1.75% | 460 | 1009 | 25 | 101 | 43 | 607 | 14 |

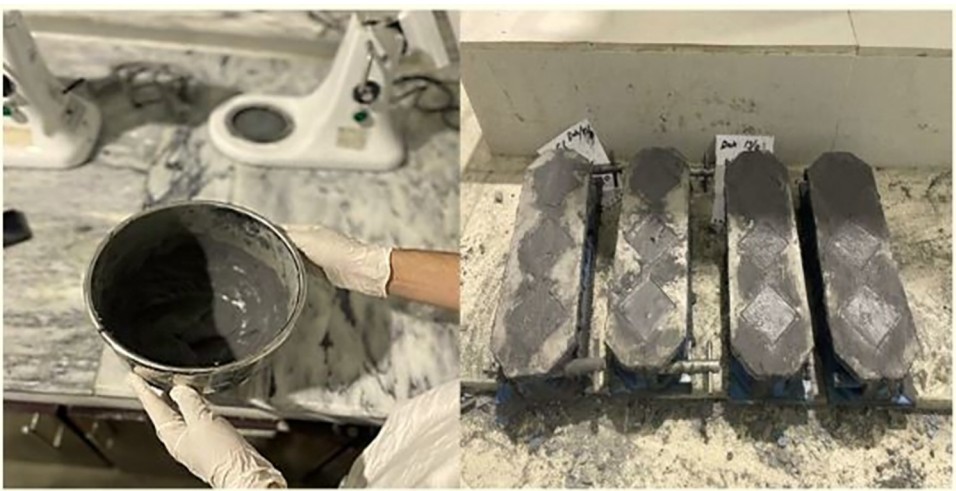

**Fig 3. Mixing and molding of AAP 50 mm cubes.**

For compression, 50 mm AAP cubes were cast. Using a universal testing machine (UTM), the test was conducted under load—controlled mode and the loading rate was set at 0.25 MPa/s [55, 56]. The mixing and molding of AAP samples is shown in Fig 3.

For uniaxial tensile testing, coupon samples of dimensions 15x40x200 mm were cast. The test was performed in accordance with ASTM D 3039/D 3039M [57].

## Workability test

Workability of the AAP mix was assessed using the flow table method outlined in ASTM C1437. Flow diameter was taken as the average reading of two perpendicular diameters on the flow table.

## Water absorption test

Water absorption in concrete/mortar impacts durability and strength. To determine the amount of water absorbed by the AAP at various hardened states, a water absorption test was conducted following the guidelines outlined in ASTM D5229. At the ages of 7th, 14th, and 28th days, samples were oven-dried for 24 hours, and their weights were recorded. Subsequently, the samples were immersed in water for 48 hours, after which their weights were measured again. The absorbed moisture content was calculated by determining the difference between the weights of the oven-dried and fully wet samples.

## Ultrasonic pulse velocity (UPV)

UPV testing, following ASTM C597–09, was conducted at 7, 14, and 28 days. Using two transducers and a pulse-receiver unit with built-in data acquisition, pulse arrival time across samples was measured. Petroleum jelly facilitated transducer-sample coupling. UPV values were calculated by dividing path length by pulse arrival time.

## Cracked OPC concrete cubes

A total of 31 OPC 150 mm cubes were cast for the experiment. This included 3 controlled specimens (OPC-CM) and 28 cracked specimens with full-length (OPC-F) and half-length (OPC-H) cracks, each with thicknesses of 1.5 mm and 3 mm, penetrating 50 mm inside the

**Table 5. Dimensions of cracks.**

| Cubes ID. | Crack length (mm) | Crack width (mm) | Crack depth (mm) |
|---|---|---|---|
| OPC-CM | – | – | – |
| OPC-1.5F | 150 | 1.5 | 50 |
| OPC-3F | 150 | 3 | 50 |
| OPC-1.5H | 75 | 1.5 | 50 |
| OPC-3F | 75 | 3 | 50 |

cube. The dimensions of the induced cracks can be found in Table 5. Grade M30 concrete, with a ratio of 1:0.75:1.5 (cement: sand: aggregate), was used for casting the concrete cubes, following the guidelines outlined in ACI 211.1–91 [58]. To induce cracks in the concrete cubes, steel plates were utilized, following the methodology established by Issa and Debs [48].

On the 28th day of wet curing, the cracks in the OPC cubes were filled by injecting AAP after thoroughly cleaning the cracked portions. The repaired samples underwent a curing process at 60°C 8for 7 days, followed by an additional 21 days of ambient temperature curing. Subsequently, the repaired samples were subjected to testing in a universal testing machine (UTM). The objective of testing was to assess the strength gain and determine whether the cracks observed at failure originated in the concrete substrate or the repaired section. The cracked cubes are depicted in Fig 4.

## Results and discussion

### Flowability test

As shown in Fig 5A, all mixes achieved a flow value greater than 145 mm which helps to assure that the mixtures were cohesive and workable. The spherical morphology of fly ash enhances the workability of AAP through what is commonly referred to as the "ball-bearing effect" [59]. Nevertheless, AAP mortars generally encounter workability challenges, inherent in their composition, compared to OPC. This challenge arises from the presence of silicates in their precursors, which possess adhesive properties. The incorporation of PE fibers exacerbates this issue, as evidenced by a noticeable decrease in flowability with increasing PE content (see Fig 5A).

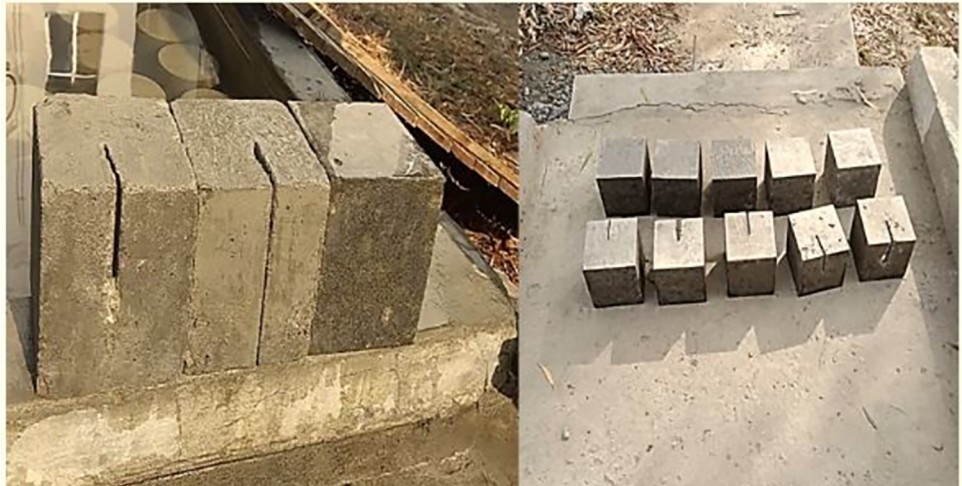

**Fig 4. Concrete cubes with cracks of different dimensions.**

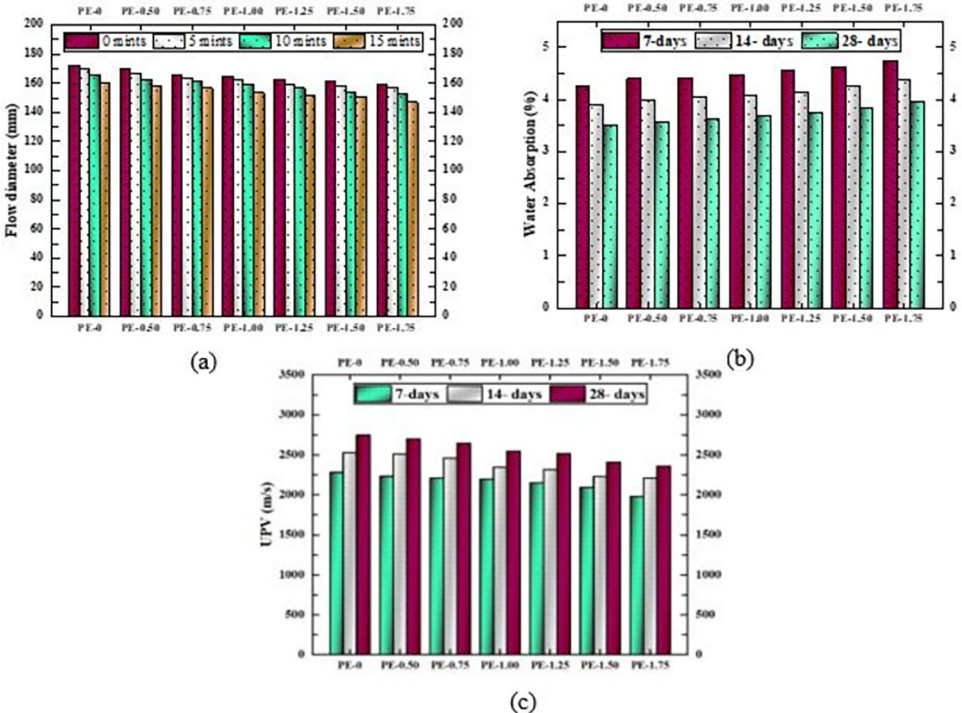

**Fig 5.** Non-destructive tests on AAP with varying percentages of PE fibers (a) Flowability test (b) Water absorption test (c) UPV test.

The decrease can be attributed to the increased pore volume around the fibers, resulting in greater water absorption and subsequently reducing the available water for sustaining workability [60]. As expected, the control mixture without PE fibers showed the highest flow diameter. Additionally, all mixtures exhibited a gradual decrease in flowability over time, as observed at intervals of 0, 5, 10, and 15 minutes.

Furthermore, Nazar et al. observed that reducing the amount of NaOH activator while maintaining $Na_2SiO_3$ constant led to a decrease in flowability. The lowest flowability was observed when $Na_2SiO_3$ was used as the sole activator without any NaOH [61, 62].

## Water absorption

A study was conducted to investigate the impact of varying PE fiber content on water absorption capacity and examine how alkali activation influences the pore structure of the AAP. Water absorption tests are performed for 7, 21, and 28 days matured samples, with each recorded value representing the average of three samples of same age (see Fig 5B). The control AAP matrix, devoid of PE fibers, exhibited the least water absorption due to alkali activation process that creates a tightly packed bond structure, thereby resisting water absorption. NaOH solution and increased activator content foster better polymerization and hydration, enhancing the matrix's resistance to water absorption [63, 64]. This resistance signifies a homogeneous dense microstructure with fewer pores and voids.

Results indicate that water absorption tends to rise with higher PE fiber content at corresponding ages, attributed to increased pore volume in the vicinity of fibers [65, 66]. For instance, the mixture containing 1.75% PE fibers displayed absorption values 11.5%, 12%, and 12.8% higher at 7, 14, and 28 days, respectively, compared to the mixture lacking PE fibers.

Additionally, water absorption decreases with age, indicating matrix development and pore reduction over time. On 28 days, water absorption decreased by 17.6% and 16.5% for samples with 0% and 1.75% PE fibers, respectively, from the seventh day values. Lower water absorption of AAP with no fibers at 7 days suggests faster reaction kinetics, implying most pores were filled with alkali activated matrix, leading to reduced water absorption.

## Ultrasonic pulse velocity (UPV)

Fig 5C illustrates the average UPV values of three samples of AAP formulation. It was observed that the AAP exhibits a superior and denser microstructure, facilitating a solid path for ultrasonic waves. UPV values for all formulations increased with curing age, indicating progressive internal structural development. At 28 days, the corresponding increase of 16 to 20%, from the 7[th] day value, in UPV indicates strengthening over time.

The velocity of ultrasonic waves was notably higher in AAP samples lacking PE fibers. Introducing 1.75% PE fibers decreases UPV values by approximately 13–14% on 7[th], 14[th] and 28[th] days. The results suggest that higher PE fiber content significantly increases internal pores and decreases the density of the AAP. Optimal PE fiber content falls within the range of 1 to 1.25%, striking a balance between internal structure, density, and flexural properties of the matrix.

## Compressive strength

The impact of PE fibers on the compressive strength of AAP is marginal and is influenced by positive fiber bridging effects [67] and the negative air entrapping effect [68]. The stress-strain graph in Fig 6 illustrates that increasing the PE fiber content up to 1% enhances the compressive strength of AAP which can be attributed to the fiber bridging effect that improves resistance against microcrack sliding and propagation by enhancing the bonding strength between the fiber and the matrix, ultimately increasing load-carrying capacity [69]. This effect is further accentuated by ensuring good dispersion of fibers throughout the matrix. Jia-Qi Wu et al. and Shaikh et al. also noted that the compressive strength of ambient cured alkali activated concrete (AAC) exhibits an increase as the volume percentage of PE fibers rises from 0.5% to 1% [70]. However, exceeding 1% PE fiber dosage leads to a reduction in compressive strength due to increased air entrainment and decreased material density. This is further exacerbated by poor fiber dispersion at higher volume fractions, increasing the likelihood of fiber balling and

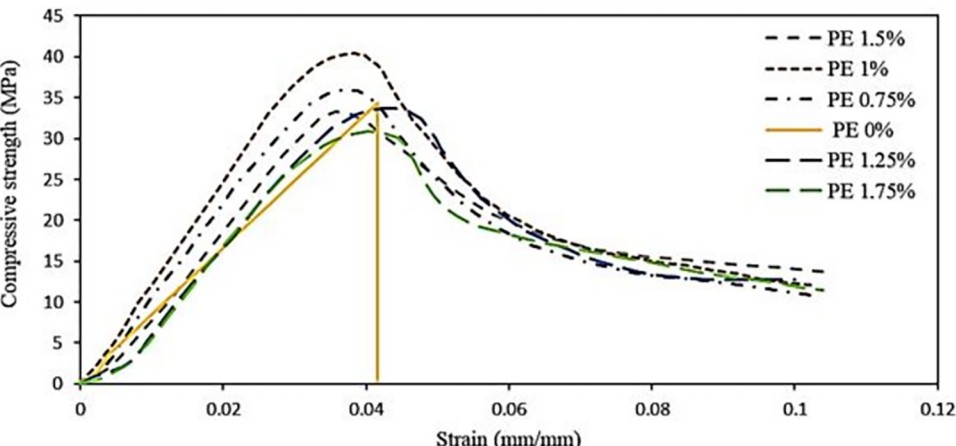

**Fig 6. Compressive strength of AAP cubes for different fibers %.**

void entrainment during mixing [45, 46]. The diminishing compressive strength with higher fiber percentages is also observed in AACs reinforced with cotton fibers [71], and a similar detrimental effect on compressive strength is noted with the incorporation of PVA fibers at volume fractions of 1.5% and 2%. Polypropylene (PP) fiber reinforced composites are not different [69, 72, 73].

Moreover, the impact of fiber on the 28-day compressive strength depends on fiber length. In AAPs with 12 mm PE fibers, as in our case, enhanced strength up to some fiber content is primarily due to the fiber-bridging effect dominating air entrainment. Conversely, as the case with 18 mm PE fibers, air entrainment prevails, leading to reduced compressive strength at all fiber percentages [50].

Notably, due to fiber addition, the failure mode of 50 mm cubes under compression is changed from cracking to squeezing, increasing the strain induced at failure [51]. Although the cube specimens experienced fracture, they retained their general form, allowing for the application of compression force. This is demonstrated by the softening tail observed in the compressive stress-strain behavior of different PE fiber reinforced-AAPs, as depicted in Fig 5. There is an abrupt failure for 0% PE fiber with little strain, while all the other fiber-reinforced AAPs have significant strain values which is helpful in applications where dynamic load resistance, seismic performance, and impact resistance are important considerations.

## Tensile strength

The uniaxial tensile strength of AAPs exhibits an increase with increasing fiber percentages, reaching a peak at 1.25% as illustrated in Fig 7. Notably, the AAP without PE fibers experiences a catastrophic failure under tension, precluding the capture of its stress-strain relationship. The stress strain graph under tension reveals zig-zag shapes indicative of cracks in multiple planes rather than a single plane, a characteristic feature of fiber reinforced AAP [74]. At a fiber percentage of 1.25%, the paste material demonstrates the maximum number of crack planes showing the material enhanced ability to resist tensile stresses. However, beyond the 1.25% PE fiber threshold, a decrease in tensile strength occurs, which is attributed to the formation of fiber clusters. Nonuniform fiber dispersion, characterized by the presence of these clusters, diminishes the effectiveness of fiber reinforcement, creating weak spots susceptible to splitting. This phenomenon, consistent with findings in related research [7, 75, 76], adversely

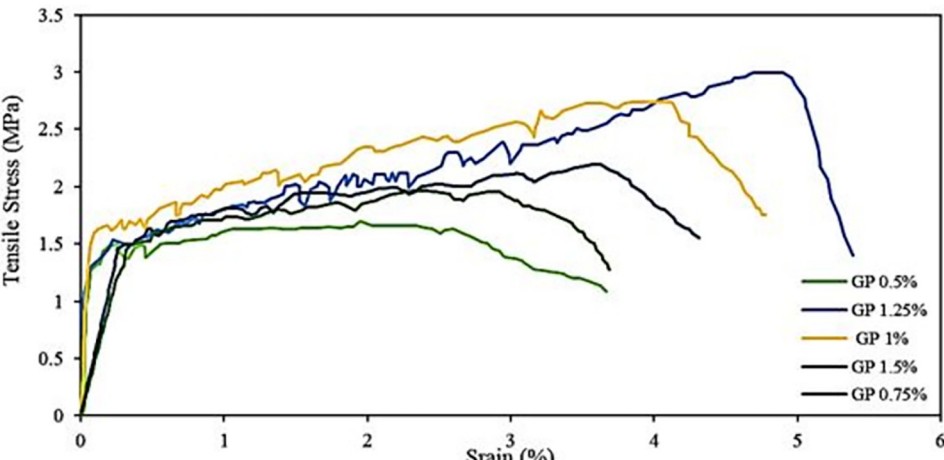

**Fig 7. Tensile strength of AAP for different PE fibers %.**

affects the overall performance of the paste. The uniaxial tensile performance of the AAPs developed in this study is consistent with the characteristics observed in fly ash-based composites studied by Ohno and Li, as well as with conventional PVA reinforced ECC [77]. From Fig 7, it is evident that in AAPs the ultimate tensile strength surpasses the initial crack strength by a significant margin. This suggests that the stress-based criterion for pseudo-strain hardening (PSH) behavior has been met [78].

## Flexural strength

Incorporating fibers into AAP enhances flexural and tensile performance by shifting the failure modes from brittle to ductile. Fibers play a crucial role by withstanding tensile stress and transforming it into shear stress at the interface between the fiber and matrix. This interaction, as illustrated in the scanning electron microscopy (SEM) image in Fig 9, creates a distinct annular region within the AAB matrix. [20, 79]. The flexural strength of AAPs increases with increasing fiber percentages, peaking at 1.25% of PE fibers (see Fig 8). This aligns with findings reported by Faiz Uddin et al. in PE-reinforced composites [80]. The graph's zig-zag shapes correspond to cracks occurring in multiple planes, a phenomenon discussed earlier. The ultimate flexural strength (modulus of rupture) surpasses the initial crack strength for all fiber reinforced samples. Moreover, the deflection at peak load exceeds that at the first crack load, indicating a consistent deflection hardening behavior in three-point bending for all samples, regardless of PE fiber content.

Research investigating the impact of PE fibers on deflection hardening of alkali activated composite (AAC) affirms that the ultimate deflection capacity of all composites rises with the PE fiber volume fraction, peaking at 0.75–1.25% [80]. However, beyond this optimal fiber volume, a decline in flexural strength, as observed in our study, occurs. The decrease is attributed to the non-uniform distribution of fibers, leading to clumping and the introduction of voids in the matrix, paralleling with the observations made in the tensile case [7].

Micro-cracks can be seen in different regions of the fragmented EGC matrix as shown in the SEM images taken by Zeya li et al. This is presumably due to excessive PE fiber-EGC matrix bonding, leading to matrix destruction. Additionally, a significant amount of paste fragments adhered to the surface of the PE fibers, indicating strong bonding between the fibers and the EGC matrix. Fig 9(E) shows debris of PE fibers within the fractured EGC matrix, possibly resulting from the firm bonding between the PE fibers and the matrix, which causes the

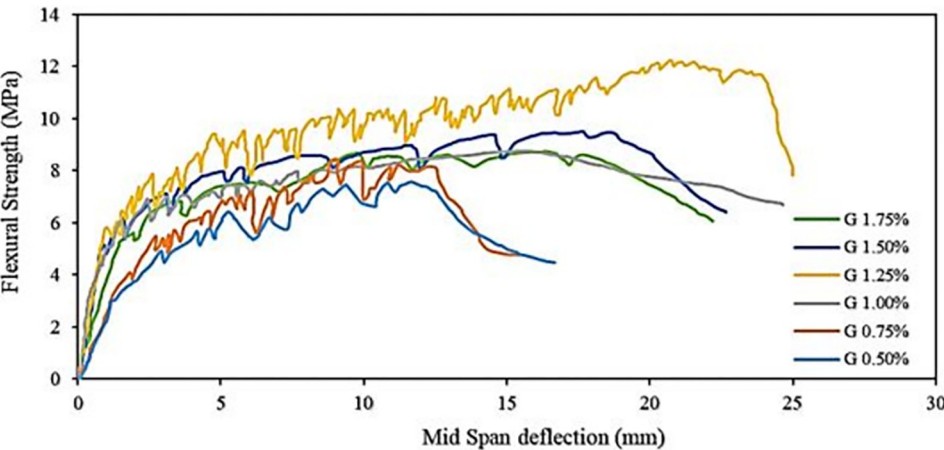

**Fig 8. Flexural strength of AAP rectangular beams for different fiber %.**

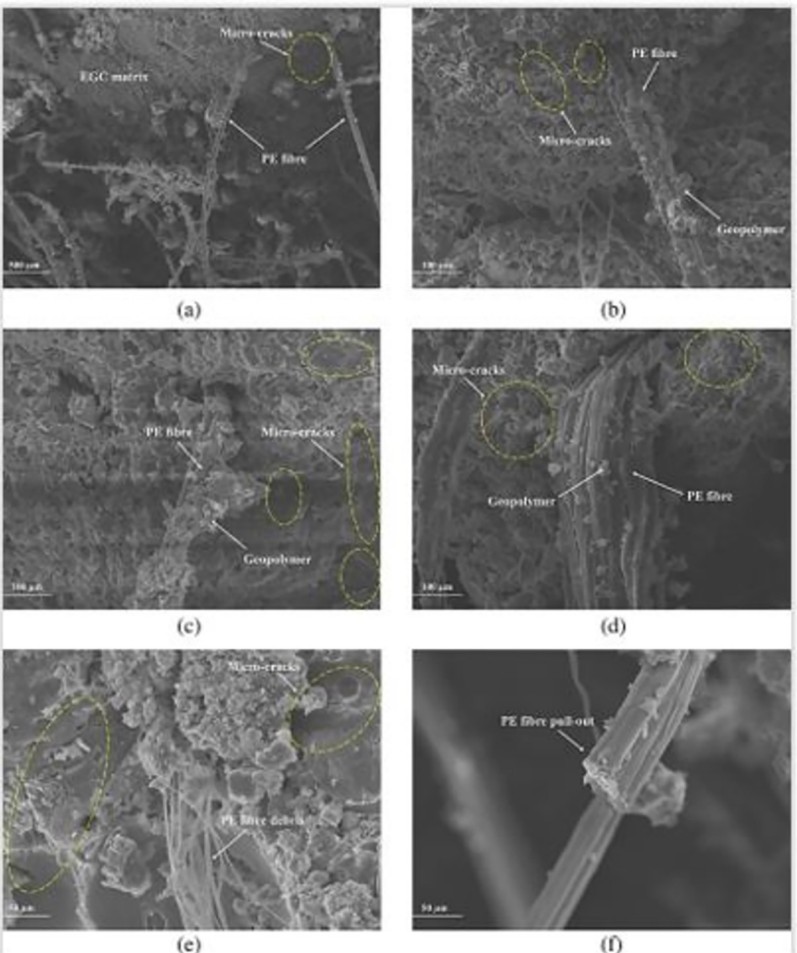

**Fig 9.** SEM images of engineered geopolymer composite (EGC), another class of AACs, containing 1.5% PE fibers: (a) Fragmented EGC matrix, (b-d) fiber interface (e) fiber debris, and fiber end. Reprinted from [69] under a CC BY license, with permission from John Wiley and sons, original copyright [2023].

fibers to chip and consume substantial friction energy. However, due to the fibers' higher nominal strength (2580 MPa), the broken EGC matrix did not break PE fibers. The cross-section of the PE fiber is also maintained in its original shape [69].

## Compressive strength of repaired cubes

Fig 10 provides a comparative analysis of the compressive strength of OPC-1.5H and OPC-3H cubes, both in their unrepaired and repaired states. Initially, unrepaired OPC-1.5H and OPC-3H cubes exhibit strengths of 22 MPa and 19.4 MPa, respectively, representing 73.3% and 64.66% of the OPC-CM's strength. Following repair with AAP, the cube strengths increase to 25.6–29.6 MPa, recovering 85.33%-98.66% of the OPC-CM strength. This signifies a substantial improvement in strength due to the repair process. Interestingly, varying the percentage of PE fibers in the repair material does not significantly impact the compressive strength of the repaired cubes. This suggests flexibility in utilizing AAP without or with any fiber percentages for effective concrete crack repair.

Notably, concrete cubes with a 1.5 mm crack exhibit higher strength compared to those with a 3 mm crack. The narrower crack width indicates a more localized and controlled

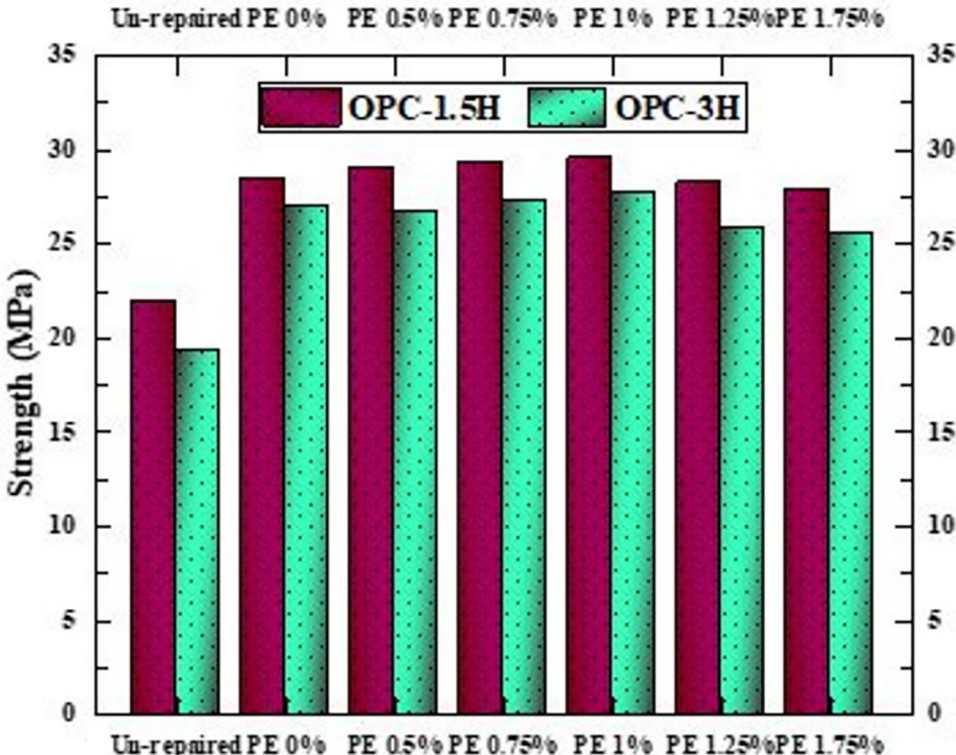

**Fig 10. Compressive strength of OPC-1.5H and OPC-3H repaired cubes.**

damage area, resulting in improved load-bearing capacity and higher strength post-repair. Furthermore, cube failure is due to the crushing of the substrate material, highlighting the integrity of the repaired surface even during failure. Similar cracking behavior were reported by B. J. Frasson et al., who used a combination of geopolymer cement paste and epoxy adhesive for repair thereby reducing a 13% drop in compressive strength of unrepaired concrete to a 3.7% for repaired concrete [81]. B. S. Meenakshi et al. [10] performed a split tensile test on the samples repaired with alkali activated mortar and OPC mortar, achieving 44% higher strength for the earlier type in comparison with the later. C. Frangieh et al. employed two distinct geopolymers, namely French and Lebanese variants, for concrete crack repair. Both categories of geopolymers exhibited superior compressive strength as compared to OPC mortar [82].

In Fig 11, a comparison of the compressive strength of OPC-1.5F and OPC-3F cubes is presented. Initially, before repair, OPC-1.5F and OPC-3F cubes exhibit strengths of 18 MPa and 15.4 MPa, corresponding to 60% and 51.3% of the OPC-CM strength. Following repair, the cubes recover 79.4%-95.4% of the OPC-CM strength. Like the case of half-length cracks, varying the percentage of PE fibers in the repair material has minimal impact on the strength of the repaired cubes. Additionally, as observed in the previous case, 1.5 mm cracks result in higher strength compared to 3 mm cracks.

For both OPC-1.5F and OPC-3F repaired cracks, cube failure is caused by substrate crushing, indicating intact repaired area even during failure, underscoring the effectiveness of AAPs in repairing concrete cracks. This observation aligns with the findings of Ding, Cheng, and Dai, who utilized AAB to fill inclined cracks in a concrete substrate. Just as our case, their examination revealed rupture within the concrete [83]. Ueng et al. performed a comprehensive investigation on the adhesion properties between mortar substrates and alkali activated cement

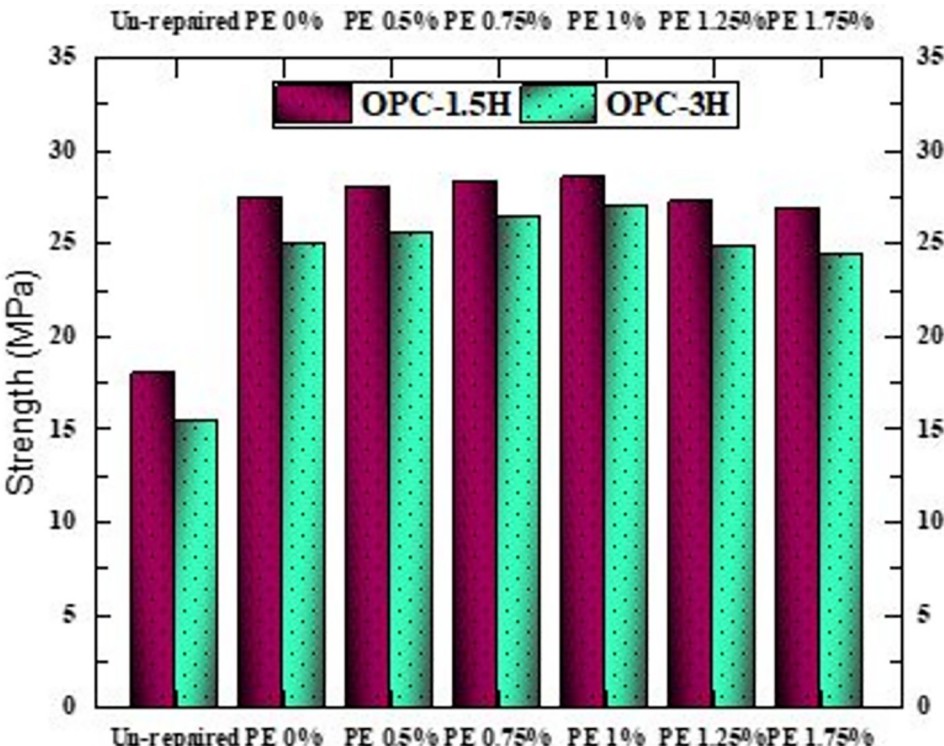

**Fig 11. Compressive strength of OPC-1.5F and OPC-3F repaired cubes.**

synthesized using metakaolin. They concluded that through an in-depth understanding of mechanical models, it is possible to make accurate predictions regarding the type of rupture and various stresses that arise in concrete when AAPs is utilized as a material for structural repair [84].

## Economic and environmental assessment

Substituting supplementary cementitious materials like fly ash for OPC proves to be a highly effective and cost-efficient strategy for reducing natural resource usage and greenhouse gas emissions, while also enhancing mechanical properties. Despite the relatively higher cost associated with the inclusion of PE fibers, the mechanical advantages they provide justify this expense. Table 6 shows the cost per kilogram in Pakistani rupees (PKR) for various raw materials employed in this study. In Table 7, it's evident that as the PE fiber content increases, the overall cost also rises. Notably, the highest cost, 63.2% greater than AAP without fibers, is observed for the mix with 1.75% PE fibers. This increment in cost is deemed acceptable when prioritizing the improvement of mechanical properties.

Emission factors, utilized to assess $CO_2$ emissions from AAP mixtures via Eq 1 [85], were derived from literature sources for individual components, as detailed in Table 8. Notably,

**Table 6. Unit cost of raw materials for AAP.**

| Raw materials | Cement | GGBS | Fly ash | $Na_2SiO_3$ | NaOH | PE fibers |
|---|---|---|---|---|---|---|
| Cost (PKR/kg) | 30 | 10 | 7 | 130 | 400 | 1100 |

**Table 7. Cost analysis of different paste mixtures.**

| Sample I.D. | Total cost (PKR/m$^3$) | % increase in price from AAP |
|---|---|---|
| PE | 26.0 | – |
| PE 0.50% | 30.7 | 18.0 |
| PE 0.75% | 33.3 | 28.1 |
| PE 1.00% | 35.3 | 35.7 |
| PE 1.25% | 37.9 | 45.7 |
| PE 1.50% | 39.8 | 53.2 |
| PE 1.75% | 42.4 | 63.2 |

cement exhibits significantly higher $CO_2$ emission factors compared to GGBS and fly ash, with the latter's emissions largely attributed to transportation. Consequently, the integration of fly ash and GGBS in concrete presents a multifaceted advantage to the construction industry, encompassing technical, economic, and sustainability considerations.

$$EF_{CONCRETE} = \sum w \times EF + EF_{MIXER} \qquad \text{Eq1}$$

## Conclusions

The mechanical properties of alkali activated paste (AAPs) depend on the percentage of polyethylene (PE) fibers used.

- Both flexural and uniaxial tensile strength of AAPs rise with increasing fiber content, reaching a maximum at 1.25% PE fibers. Compared to 0.5% PE fiber reinforced AAPs, those with 1.25% PE fibers exhibit 100% greater tensile strength and 70% greater flexural strength. However, strength decreases beyond this optimal fiber percentage. Notably, all fiber-reinforced pastes displayed strain-hardening behavior, regardless of the PE fiber content.

- The compressive strength of AAPs mirrors the trend observed in flexural and tensile strength and is maximum for 1% PE fibers, surpassing AAP without fibers by 17%. Beyond this fiber percentage, the compressive strength decreases. A stress-strain softening tail is observed for all fiber reinforced AAPs under compression.

- In compression, due to the addition of fibers, the failure mode is changed from abrupt cracking to squeezing while in the case of flexure and tension the substrate failed after cracking in multiple planes.

  AAP are viable filler materials for concrete crack repair.

- OPC-3F and OPC-1.5F cracked cubes exhibited only 51.3% and 60% of OPC-CM strength respectively. However, when these cubes are repaired, depending on the width of the crack,

**Table 8. $CO_2$ emissions for different mix components.**

| Concrete component | EF (Kg-CO$_2$/ton) |
|---|---|
| OPC [85] | 650 |
| GGBS [86] | 182 |
| Fly ash [87] | 0.023 |
| NaOH [88] | 1915 |
| Na$_2$SiO$_3$ [88] | 1514 |
| Paste mixer [85] | 1.61 |

they regain 79.4% to 95.4% of their strength. The same is the case for OPC-1.5H and OPC-3H, which regained 85.33% to 98.66% of the OPC-CM strength. Thus, the use of AAP as fillers can greatly mitigate the strength loss associated with cracks in concrete.

- When used as a filler material, the proportion of fibers in AAP has no substantial effect on the compressive strength of the repaired cubes. Consequently, there is more flexibility in using AAP with any fiber percentage while still achieving effective concrete crack repair.

## Recommendations

Here are some recommendations for future research:

- Ascertain the long-term durability of cracks repaired with AAP.

- Examine how fiber-reinforced pastes, used as repair materials, affect the performance of repaired cubes under flexural and tensile loads.

- Investigate different crack geometries to encompass different types of cracks.

- Investigate the effect of various curing conditions on the effectiveness of crack repair.

By addressing these research areas, a thorough grasp of material selection and crack repair methods can be achieved, enhancing the overall durability of repaired structures.

## Author Contributions

**Conceptualization:** Munir Iqbal.

**Funding acquisition:** Loai Alkhattabi, Hisham Alabduljabbar.

**Investigation:** Munir Iqbal, Sohaib Nazar, Loai Alkhattabi, Jihad Alam, Hisham Alabduljabbar.

**Methodology:** Munir Iqbal, Loai Alkhattabi, Hisham Alabduljabbar.

**Project administration:** Muhammad Ashraf.

**Supervision:** Muhammad Ashraf.

**Validation:** Munir Iqbal, Loai Alkhattabi, Jihad Alam, Hisham Alabduljabbar, Zahoor Khan.

**Writing – original draft:** Munir Iqbal.

**Writing – review & editing:** Muhammad Ashraf, Sohaib Nazar.

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
