## [Decision Letter · Decision Letter 0]

16 Apr 2024

PONE-D-24-09148Innovative alkali activated paste incorporated with polyethylene fibers for concrete crack repair: a comprehensive development studyPLOS ONE

Dear Dr. Iqbal,

Thank you for submitting your manuscript to PLOS ONE. After careful consideration, we feel that it has merit but does not fully meet PLOS ONE’s publication criteria as it currently stands. Therefore, we invite you to submit a revised version of the manuscript that addresses the points raised during the review process.

We look forward to receiving your revised manuscript.

Kind regards,

Parthiban Kathirvel

Academic Editor

PLOS ONE

Journal Requirements:

3. We note that Figures 2 and 3 in your submission contain copyrighted images. All PLOS content is published under the Creative Commons Attribution License (CC BY 4.0), which means that the manuscript, images, and Supporting Information files will be freely available online, and any third party is permitted to access, download, copy, distribute, and use these materials in any way, even commercially, with proper attribution. For more information, see our copyright guidelines: http://journals.plos.org/plosone/s/licenses-and-copyright.

a. You may seek permission from the original copyright holder of Figures 2 and 3 to publish the content specifically under the CC BY 4.0 license. 

Reviewers' comments:

Reviewer's Responses to Questions

**Comments to the Author**

1. Is the manuscript technically sound, and do the data support the conclusions?

Reviewer #1: Yes

Reviewer #2: Yes

2. Has the statistical analysis been performed appropriately and rigorously? 

Reviewer #1: N/A

Reviewer #2: Yes

3. Have the authors made all data underlying the findings in their manuscript fully available?

Reviewer #1: Yes

Reviewer #2: Yes

4. Is the manuscript presented in an intelligible fashion and written in standard English?

Reviewer #1: No

Reviewer #2: Yes

5. Review Comments to the Author

Reviewer #1: Interesting topic. However, authors need to describe the novelty of the word since there similar paper on the topic.

Improve title… Innovative alkali activated paste [[incorporated with???]] polyethylene fibers for 2 concrete crack repair: [[a comprehensive development study???]].

Improve… This study investigates the development of [[one part???]] alkali activated paste (AAP) with its application 13 in repairing concrete cracks

Short sentence… Basalt fibers increase compressive strength [14]

Sketches about repair methods can be clarified further.

Referencing style… Rodler and Fulton [37,38????] outline gravity filling as a 4 79 method employing low viscosity resins or monomers like urethanes and epoxies for repairing narrow 80 cracks (0.03 mm to 2 mm) [37,38]???

B. S. Meenakshi et al. ??? [7] performed a split tensile test on the repaired samples and achieved 44% higher strength for alkali activated mortar in comparison to conventional mortar [7]??.

Captions (e.g. Figure can be improved)

Conclusions can be improved. E.g. adding values on percentage trend.

Details are not necessary…

UPV measurements were conducted by securely coupling the transducers to opposite ends of the specimens, with petroleum jelly serving as the coupling agent between the transducer and the specimen. The UPV test involved recording the pulse arrival time, which denotes the duration between the application of the pulse and its arrival on the opposite facet of the specimen. Subsequently, the UPV value was calculated by dividing the path length by the pulse arrival time.

English (see also above)

Despite causing a reduction in slump and [[introducing challenges??]] such as uneven fiber distribution, PE fibers improve abrasion resistance and significantly decrease drying shrinkage by over 10% [23–26].

These included [[coupon???] samples for tensile testing, rectangular beams for flexural testing, and 50 mm cubes for compression testing.

Reviewer #2: 1) The Abstract should be enriched with the brief details of the methodology.

2) The problem to be addressed in this study should also be highlighted in the Abstract.

3) Please highlight the novelty in the Abstract also.

4)The novelty and significance of the present work should be highlighted in the last paragraph of the Introduction section.

5)There are no critical review/discussions before the Conclusions. Authors should add it.

6) The authors are recommended to add latest relevant literature review on such works

Akash Kanna, G., Parthasarathi, N. (2024). Experimental Investigation on Concrete by Partial Replacement of Fine Aggregate with Olivine Sand. In: Gencel, O., Balasubramanian, M., Palanisamy, T. (eds) Sustainable Innovations in Construction Management. ICC IDEA 2023. Lecture Notes in Civil Engineering, vol 388. Springer, Singapore. https://doi.org/10.1007/978-981-99-6233-4_6

Parthasarathi Narayanaswamy, K.S. Satyanarayanan M. Prakash. Developments and research on fire-induced progressive collapse behaviour of reinforced concrete elements and frame - a review. Environ Sci Pollut Res Int. 2022 Aug 3. doi: 10.1007/s11356-022-22336-x.

N.Parthasarathi, K. S. Satyanarayanan, M.Prakash, Ehsan Noroozinejad Farsangi, V.Thirumurugan, S. Srinivasa Senthil (2022),Progressive Collapse evaluation of RC frames under high temperature conditions: Experimental and Finite element investigations, Structures- Elseiver, Volume 41, Pages 375-388. https://doi.org/10.1016/j.istruc.2022.05.037

6. PLOS authors have the option to publish the peer review history of their article (what does this mean?). If published, this will include your full peer review and any attached files.

Reviewer #1: No

Reviewer #2: No

---

## [Author Response · Author response to Decision Letter 0]

13 May 2024

REVIEWER #1

1) Interesting topic. However, authors need to describe the novelty of the word since there are similar paper on the topic.

Response: Thank you for taking the time to review our manuscript and for providing valuable feedback. We are grateful for your recognition of the intriguing topic explored in our study.

In response to your comment regarding the novelty of our work, we have emphasized in the second last paragraph of the introduction section that while there are existing papers on crack repair utilizing various filler materials, our study introduces a novel approach by employing sustainable repair materials such as alkali activator paste (AAP). Still there exist few articles that have explored the use of sustainable binders too, but they have often focused on different crack geometries and haven’t delved into the mechanical behavior of the repair material. Lastly, the incorporation of polyethylene (PE) fibers into the AAP and investigating its development and viability within filler material context adds a unique dimension to our research, distinguishing it from previous works in the field. 

We have ensured that the novelty of our approach is clearly articulated in the final paragraph of the introduction section. 

2) Improve title… Innovative alkali activated paste [[incorporated with???]] polyethylene fibers for 2 concrete crack repairs: [[a comprehensive development study???]].

 Response: Thank you for your feedback. We have carefully considered your suggestion and updated the title to better reflect the content and focus of our study. The new title now reads: "Development of innovative alkali activated paste reinforced with polyethylene fibers for concrete crack repair." We believe that this revised title accurately encapsulates the key elements of our research, highlighting the development of alkali activated paste and its reinforcement with polyethylene fibers for the purpose of concrete crack repair.

3)Improve… This study investigates the development of [[one part???]] Alkali activated paste (AAP) with its application 13 in repairing concrete cracks.

Response: Thank you for your constructive feedback. We have considered your suggestion for improvement regarding the description of our study's focus. 

In response, we have revised the sentence to better convey the aim and scope of our research. The updated sentence now reads: "This study aims to develop alkali activated paste (AAP) and investigates its efficiency in repairing concrete cracks." We have highlighted this modification in red within the manuscript for easy identification.

4) Short sentence… Basalt fibers increase compressive strength [14]. 

Response: Regarding the specific comment about the shortness of the sentence on basalt fibers, we have considered your suggestion and combined the sentence with another sentence and now it reads as follows: "Basalt fiber [14] increases compressive strength while steel fibers improve tensile strength and provide good freeze-thaw resistance."

We believe that this modification strengthens the overall flow.

5) Sketches about repair methods can be clarified further.

Response: Regarding the comment about clarifying the sketches on repair methods, we have taken your suggestion. We have diligently revised Figure 3 to ensure a clearer depiction. Additionally, we have adjusted the sizes of some images to enhance their clarity and vividness. We hope these modifications adequately address the raised concerns and contribute positively to the overall quality of the manuscript. 

6) Referencing style… Rodler and Fulton [37,38????] outline gravity filling as a 4 79 method employing low viscosity resins or monomers like urethanes and epoxies for repairing narrow 80 cracks (0.03 mm to 2 mm) [37,38]???

B. S. Meenakshi et al. ??? [7] performed a split tensile test on the repaired samples and achieved 44% higher strength for alkali activated mortar in comparison to conventional mortar [7]??.

Response: Your attention to details has greatly contributed to the refinement of our work. Regarding the comment on referencing, we have carefully considered your suggestion and made the necessary adjustments. We have repositioned the references directly after the authors' names to enhance the clarity of citations. Once again, we appreciate your valuable input and guidance throughout this review process.

7) Captions (e.g. Figure can be improved)

Response: As already discussed, we have diligently revised Figure 3 to ensure a clearer depiction. Additionally, we have adjusted the sizes of some images to enhance their clarity and vividness. Moreover, ambiguous sentences in the whole manuscript, including those in the captions, have been improved. We hope these modifications adequately address the raised concerns.

8) Conclusions can be improved. E.g. adding values on percentage trend.

Response: Your insights have been instrumental in refining our conclusions and strengthening the overall impact of our work. As per your suggestions, we have added values in percentage trends to enhance the clarity and depth of our findings. These adjustments are highlighted in red within the revised manuscript. We are confident that these enhancements will enrich the conclusions.

9) Details are not necessary…

UPV measurements were conducted by securely coupling the transducers to opposite ends of the specimens, with petroleum jelly serving as the coupling agent between the transducer and the specimen. The UPV test involved recording the pulse arrival time, which denotes the duration between the application of the pulse and its arrival on the opposite facet of the specimen. Subsequently, the UPV value was calculated by dividing the path length by the pulse arrival time. Improve English (above.)

Response: Thank you for highlighting it out. Brevity can often enhance clarity and efficiency. I'm glad that the details are now presented in the UPV section in a more concise manner and with improved English. Changes are highlighted in red. 

10) Despite causing a reduction in slump and [[introducing challenges??]] such as uneven fiber distribution, PE fibers improve abrasion resistance and significantly decrease drying shrinkage by over 10% [23–26].

These included [[coupon???] samples for tensile testing, rectangular beams for flexural testing, and 50 mm cubes for compression testing.

Response: Your keen observations have greatly contributed to the refinement of our work. We acknowledge the importance of your suggestions regarding the identified inaccuracies in the sentences you highlighted. Following your guidance, we have carefully revised these sentences, ensuring accuracy and clarity. The revised sections have been appropriately highlighted in red within the introduction section of the manuscript.

REVIEWER #2:

1) The Abstract should be enriched with the brief details of the methodology.

Response: We have carefully considered the suggestion to enrich the abstract with brief details of the methodology, and we wholeheartedly agree with its merit. As per the reviewer's recommendation, we have included a succinct description of the methodology within the abstract. Specifically, we have added the following sentence to provide readers with essential insights into our experimental approach:

"To evaluate the repair performance of AAP, ordinary Portland cement (OPC) cubes were cast with pre-formed cracks. These cracks were induced by placing steel plates during casting and were designed to be full and half-length with widths of 1.5 mm and 3 mm." Thank you for guiding us in this refinement process.

2) The problem to be addressed in this study should also be highlighted in the Abstract.

Response: We sincerely appreciate your insightful feedback provided to further refine the abstract of our manuscript. Upon careful consideration of your suggestion, we have modified the abstract to ensure that the problem addressed in our study is prominently highlighted. We’ve added the following sentences in the abstract: “Concrete structures are susceptible to cracking, which can compromise their integrity and durability. Repairing them with ordinary Portland cement (OPC) paste causes shrinkage cracks to appear in the repaired surface. Alkali-activated binders offer a promising solution for repairing such cracks. This study aims to develop an alkali-activated paste (AAP) and investigate its effectiveness in repairing concrete cracks.”

We are confident that this clarification will enhance the abstract's effectiveness in conveying the problem addressed by our research. 

3) Please highlight the novelty in the Abstract also.

Response: While existing research explores various filler materials for crack repair, our study introduces a novel approach by employing a sustainable repair material – alkali-activated paste (AAP). Still, some studies have explored the use of sustainable binders too, they often focus on different crack geometries and lack investigation into the mechanical behavior of the repair material itself. Our research incorporates polyethylene (PE) fibers into the AAP, investigating its development and viability within the filler material context. This adds a unique dimension to our research, distinguishing it from previous work in the field. The novelty as discussed above, is clearly articulated in the abstract. 

4)The novelty and significance of the present work should be highlighted in the last paragraph of the Introduction section.

Response: We extend our gratitude for your insightful feedback. We have carefully revised the last paragraph of the Introduction to clearly articulate the novelty and significance of our work. We believe that this modification will effectively underscore the unique contributions and importance of our research to the scientific community.

5)There are no critical review/discussions before the Conclusions. Authors should add it.

Response: Upon careful consideration of your comment regarding the absence of critical review and discussions, we have meticulously expanded upon the results obtained in this study, providing detailed explanations supported by scientific evidence. Furthermore, we have ensured proper comparison with relevant literature to contextualize our findings within the broader scientific discourse. To facilitate clarity and comprehension, we have modified certain paragraphs in the Result and Discussion section, with the relevant changes highlighted in red. We believe that these modifications will enrich the manuscript with critical review and discussions that enhance the depth and rigor of our analysis. 

6) The authors are recommended to add latest relevant literature review on such works.

Akash Kanna, G., Parthasarathi, N. (2024). Experimental Investigation on Concrete by Partial Replacement of Fine Aggregate with Olivine Sand. In: Gencel, O., Balasubramanian, M., Palanisamy, T. (eds) Sustainable Innovations in Construction Management. ICC IDEA 2023. Lecture Notes in Civil Engineering, vol 388. Springer, Singapore. https://doi.org/10.1007/978-981-99-6233-4_6

Parthasarathi Narayanaswamy, K.S. Satyanarayanan M. Prakash. Developments and research on fire-induced progressive collapse behaviour of reinforced concrete elements and frame - a review. Environ Sci Pollut Res Int. 2022 Aug 3. doi: 10.1007/s11356-022-22336-x

N.Parthasarathi, K. S. Satyanarayanan, M.Prakash, Ehsan Noroozinejad Farsangi, V. Thirumurugan, S. Srinivasa Senthil (2022),Progressive Collapse evaluation of RC frames under high temperature conditions: Experimental and Finite element investigations, Structures- Elseiver, Volume 41, Pages 375-388. https://doi.org/10.1016/j.istruc.2022.05.037

Response: Acknowledging the importance of incorporating the latest relevant literature into our work, we have carefully considered the reviewer's recommendation. In response, we have diligently reviewed the latest pertinent literature on similar research topics. Citing the prescribed journal articles, we have incorporated relevant information into our manuscript.

Thanks again for all valuable suggestions. We are open to further suggestions and welcome any additional feedback you may have.

---

## [Decision Letter · Decision Letter 1]

27 May 2024

Development of innovative alkali activated paste reinforced with polyethylene fibers for concrete crack repair.

PONE-D-24-09148R1

Dear Dr. Iqbal,

We’re pleased to inform you that your manuscript has been judged scientifically suitable for publication and will be formally accepted for publication once it meets all outstanding technical requirements.

Kind regards,

Parthiban Kathirvel

Academic Editor

PLOS ONE

Reviewers' comments:

Reviewer's Responses to Questions

**Comments to the Author**

1. If the authors have adequately addressed your comments raised in a previous round of review and you feel that this manuscript is now acceptable for publication, you may indicate that here to bypass the “Comments to the Author” section, enter your conflict of interest statement in the “Confidential to Editor” section, and submit your "Accept" recommendation.

Reviewer #2: All comments have been addressed

2. Is the manuscript technically sound, and do the data support the conclusions?

Reviewer #2: Yes

3. Has the statistical analysis been performed appropriately and rigorously? 

Reviewer #2: N/A

4. Have the authors made all data underlying the findings in their manuscript fully available?

Reviewer #2: Yes

5. Is the manuscript presented in an intelligible fashion and written in standard English?

Reviewer #2: Yes

6. Review Comments to the Author

Reviewer #2: Authors corrected the given comments given by the reviewers, and the manuscripts can be published in the journal.

7. PLOS authors have the option to publish the peer review history of their article (what does this mean?). If published, this will include your full peer review and any attached files.

Reviewer #2: **Yes: **N. PARTHASARATHI

---

## [Editor Report · Acceptance letter]

4 Jun 2024

PONE-D-24-09148R1 

PLOS ONE

Dear Dr. Iqbal, 

I'm pleased to inform you that your manuscript has been deemed suitable for publication in PLOS ONE. Congratulations! Your manuscript is now being handed over to our production team.

Kind regards, 

on behalf of

Dr. Parthiban Kathirvel 

Academic Editor

PLOS ONE